# SiO₂ Nanoparticles as New Repairing Treatments toward the Pietraforte Sandstone in Florence Renaissance Buildings

**Federica Valentini** [1,*], **Pasquino Pallecchi** [2], **Michela Relucenti** [3], **Orlando Donfrancesco** [3], **Gianluca Sottili** [4], **Ida Pettiti** [5], **Valentina Mussi** [6], **Sara De Angelis** [1], **Claudia Scatigno** [7] and **Giulia Festa** [7]

1 Department of Sciences and Chemical Technologies, Tor Vergata University, Via della Ricerca Scientifica 1, 00133 Rome, Italy
2 Soprintendenza Archeologia Belle Arti e Paesaggio per la Città Metropolitana di Firenze e le Province di Pistoia e Prato, Piazza Pitti 1, 50100 Firenze, Italy
3 Department of Anatomical Legal Histological Sciences and of the Locomotor Apparatus, Sapienza University, Via Alfonso Borelli 50, 00161 Rome, Italy
4 Department of Earth Sciences, Sapienza University, Piazzale Aldo Moro 5, 00185 Rome, Italy
5 Department Chemistry, Sapienza University, Piazzale Aldo Moro 5, 00185 Rome, Italy
6 IMM-CNR Institute of Microelectronics and Microsystems, National Research Council Area della Ricerca di Roma Tor Vergata, Via del Fosso del Cavaliere 100, 00133 Rome, Italy
7 CREF-Museo Storico della Fisica e Centro Studi e Ricerche "Enrico Fermi", Via Panisperna 89a, c/o Piazza del Viminale 1, 00189 Rome, Italy
* Correspondence: federica.valentini@uniroma2.it

**Abstract:** In this work, the consolidation efficiency of SiO₂ nanoparticles (synthesized in the Chemistry laboratories at the Tor Vergata University of Roma) was tested on Pietraforte sandstone surfaces belonging to the bell tower of San Lorenzo (Florence, Italy) and was fully investigated. Nanoparticles (synthesized in large-scale mass production) have been characterized by XRD—X-Ray Diffraction; Raman and FTIR—Fourier Transform Infrared spectroscopy; SEM—Scanning Electron Microscopy; while the Pietraforte sandstone morphology was examined by Porosimetry, capillary absorption test, surface hardness test, drilling resistance and tensile strength. The colorimetric measurements were also performed to characterize the optical modification exhibited by Pietraforte sandstones, especially after the SiO₂ treatments. Our results show that applying to the Pietraforte, the new consolidating agent based on SiO₂ nanoparticles, has several advantages, as they are more resistant to perforation, wear, and abrasion even long range (for long times of exposure and consolidating exercise against Florentine sandstone), compared to the CaCO₃ nanoparticles (tested in our previous paper), which instead show excellent performance but only close to their first application. This means that over time, their resistance to drilling decreases, they wear much more easily (compared to SiO₂-treated sandstone), and tend to exhibit quite a significant surface abrasion phenomena. The experimental results highlight that the SiO₂ consolidation efficiency on this kind of Florentine Pietraforte sandstone (having low porosity and a specific calcitic texture) seems to be higher in terms of water penetration protection, superficial cohesion forces, and an increase in surface resistance. Comparing the performance of SiO₂ nanoparticles with commercial consolidants in solvents such as Estel 1000 (tested here), we demonstrate that: (A) the restorative effects are obtained with a consolidation time over one week, significantly shorter when compared to the times of Estel 1000, exceeding 21 days; (B) SiO₂ nanoparticles perform better than Estel 1000 in terms of cohesion forces, also ensuring excellent preservation of the optical and color properties of the parent rock (without altering it after application).

**Keywords:** (SiO₂) nanoparticles; low porosity sandstone; Pietraforte sandstone; texture; aqueous consolidation treatments; chemical synthesis; cohesion forces; water vapour resistance (μ); water vapor permeability (%); optical features; treatment efficiency (%)

## 1. Introduction

In this work, some products used for the conservation of the Pietraforte sandstone used in the construction of the main historical monuments of Florence have been analyzed. The first evidence of the use of the Pietraforte in Florence dates back to the Roman period with the construction of the theater [1], whose remains are now preserved under Palazzo Vecchio, but its main use is in the Middle Ages and the Renaissance. In these periods, the Pietraforte represents the most important building material with which the most important churches and the most important historical Florentine palaces are built, including Palazzo Vecchio (XIIIth century), Orsanmichele (XIVth century), Palazzo Medici-Riccardi (XIVth century), Palazzo Pitti (XVIth century) the church of S. Remigio (XIVth century) and that of San Lorenzo (XVth century).

Florentine Pietraforte sandstone is a turbidite sedimentary rock belonging to the homonymous formation of the Calvana supergroup (external Ligurides). Its typical color changes from bluish gray to brown, with veins of calcite; it tends to ocher shades due to the chemical reaction of the iron oxides. Its geological age dates back to the upper Cretaceous period, that is to say, about 150 million years ago. The Pietraforte is classified from a petrographic point of view as a lithic sandstone characterized by a clastic component essentially of quartz, feldspar, calcite, dolomite, and fragments of sedimentary, metamorphic, and effusive rocks. The matrix consists of micritic calcite with a small number of clay minerals and a secondary calcite cement [2], which have been highlighted in Table 1.

**Table 1.** Summary list of the properties of the Pietraforte sandstone.

| Materials | Lithological Type | Geological Formation | Age | Historic Quarries | Petrographic Classification | Distinguishing Features |
|---|---|---|---|---|---|---|
| Pietraforte sandstone | Sedimentary rock | The Pietraforte Formation is an allochthonous unit of the Ligurian domain in the northern Apennines | Cretaceous Superior (90–70 Ma) | Reliefs on the left bank of the Arno river near Florence (Costa San Giorgio, Boboli, Bello sguardo Monte Ripaldi) | Fine grains lithic sandstone | - Porosity 4–6%;<br>- Breaking load 140 MPa<br>- Plane-parallel and convoluted lamination<br>- Calcite veins |

In Figure 1, a thin section micrograph of the Pietraforte sandstone includes the clastic fraction consisting of quartz, micas, and dolomitic rock fragments and the matrix consisting of micritic calcite (optical transmitted light microscopy, xpl).

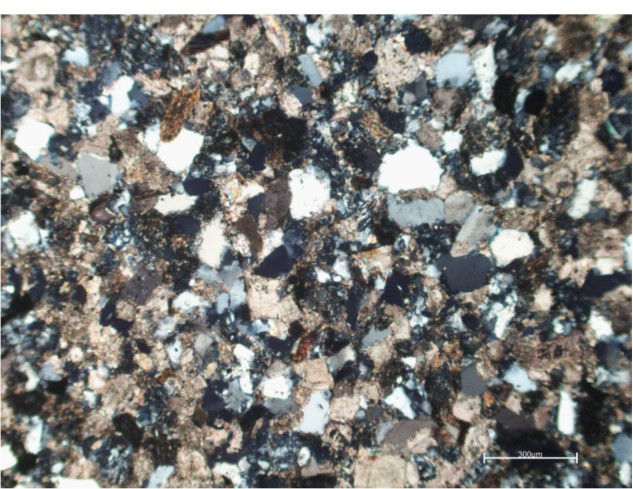

**Figure 1.** Thin section of a sample of Pietraforte, collected in the bell tower of San Lorenzo (Florence).

In Table 2, the main technical features concerning the Pietraforte sandstones have been summarized.

**Table 2.** Technical features of Pietraforte sandstones.

| | |
|---|---|
| Absorption of water at atmospheric pressure | 0.80% |
| Apparent density | 2.74 Kg/m$^3$ |
| Open porosity | 1.40% |
| Flexural strength with a concentrated load | 32 MPa |
| Compressive strength | 198 MPa |
| Slip resistance | 38 USRV |
| Freezing UNI6506 | did not freeze |

Their degradation results from a natural transformation related to both the intrinsic properties of the rock itself (mineralogical and chemical composition, structure and texture, physical characteristics) and the external environment in which it is inserted. In general, deterioration always occurs when the rock is exposed to atmospheric agents, such as water, temperature variations, pollution, etc., which modify its part of the mineral components. The "History of altering processes" of a stone material begins from the moment of its extraction, an operation that provokes the development of latent fractures in the rock, causing more or less superficial microfractures that will significantly encourage the action of agents responsible for degradation.

Among the processes of degradation, the temperature variations and the freezing phenomena represent the most relevant events. In the first case, the anisotropic variations of the minerals' size occur, and in the second, the formation of ice into the porosity provokes several significant tensions.

In particular, the presence of water in the porosity of the Pietraforte induces the dissolution of the carbonate matrix and absorption of water by the clayey minerals producing arenization processes in the stone. All these actions inevitably lead to the loss of cohesion of the rock.

In many archaeological areas, historic palaces and places dedicated to the conservation of cultural heritage with sites in the region of Tuscany [3–5], ethyl silicate and its derivatives are widely used in the past and still today in the consolidation treatments of Florentine sandstone rock, typically Pietraforte, Pietra Serena and Pietra Bigia. Ethyl silicate (trade name Estel1000) is widely applied as it represents a real consolidating agent and not a protective compound nor a water-repellent material.

For example, the façades of Palazzo Strozzi [6] were first consolidated with Estel1000, and then it was protected with a fluorinate\d protective agent. Palazzo Strozzi is one of the historic residences of the Florentine Renaissance (currently located in the historic center of the city of Florence, between Strozzi Square and Via Tornabuoni), mainly made of Florentine Pietraforte and Pietra Serena sandstones (but also of several other materials as ancient mortars, etc.).

Palazzo Medici Riccardi and Palazzo Pitti in Florence were also treated with Estel1000, (by CTs Srl international company, [6]).

Another example of the application of Estel 1000 [6] as a commercially available consolidating agent was in Florence in Villa Corsini di Castello (1500–1550) on the occasion of the restoration and consolidation of the "Allegory of the river" (a renaissance sculpture), by means Niccolò Pericoli, known as Tribolo.

These silicates (which not only include Estel1000 but a wide range of commercially available products) have many advantages such as long duration and stability over time of the treatment, absence of alteration of the optical properties of the Florentine sandstone (without color changes after the treatment of the sandstone). On the other hand, such products also have non-advantageous aspects such as solvent application and spreading,

these being commercially prepared in a liquid working medium. This results in longer response times in terms of the effectiveness of the consolidating treatment. In 21 days, there was an effective consolidating action against the Florentine sandstone [7].

For this reason, conservation scientists have thought and designed new materials based on silicates, but with different properties that derive directly from the dimensions confined in the nanoscale, as the different dispersibility in several different working mediums, generating a nanosuspension and not a consolidating agent in a solvent. According to what has been said previously, several nanoparticles have been synthesized and applied to Florentine sandstones by many authors in literature, such as $SiO_2$ [8–11], $Ca(OH)_2$/nano lime [12–14], $CaCO_3$ [15,16], nanocomposite materials [17,18] and hybrid nanocomposite materials based on consolidating agents and biocide, as reported in the literature [19].

The main advantage of applying these nanodispersions concerns a faster diffusion in low porous sandstone compared to the commercially available silicate products. Therefore, it follows that the consolidation efficiency will be maximum in a shorter time (about only 1 week, [20,21]), while in the case of commercial consolidating agents, the consolidation efficiency will take place on a longer time scale ($\geq$21 days), [22,23]. This effect is mainly due to the viscosity of the solvent where the commercial consolidating reagent has been previously dissolved (the viscosity of Estel1000 is equal to 10 cP at Room Temperature, 25 °C, [6] corresponds to 100 mPa.s.

The viscosity of the nanodispersions reported in the literature with consolidating application (exclusively on Florentine Pietraforte) is negligible compared to that of commercially silicate reagents, thus justifying the better diffusion of nanodispersions in times certainly shorter than those exhibited by commercially available consolidants. For this purpose, some typical viscosity values of nanodispersions applied directly on the Florentine Pietraforte, described in the literature [24–28], are summarized in Table S2, see Supplementary Materials.

All these solvents are compatible with the substrate of Pietraforte, and Pietra Serena, as reported in the technical specifications of the Central Institutes of Rome (www. icr.beniculturali.it), but the diffusion coefficients vary significantly with the viscosity of the medium and also with the degree of functionalization of the synthesized nanoparticles. In general, the diffusion coefficient decreases as the viscosity and the degree of functionalization/branching of the nanoparticles (both) increase (see the Table S3 see Supplementary Materials).

For this reason, with this work, the authors thought of designing and manufacturing new $SiO_2$ nanoparticles (without functionalization) that were dispersed in an aqueous working medium, above all compatible with the Florentine sandstone substrates and also with end users. These innovative particles are synthesized with an eco-sustainable, low environmental impact Green Chemistry technique (in a purely aqueous solution) with a quantitative yield towards scalable (mass) production. The application of these nanoparticles to the Pietraforte sampled at the bill of the San Lorenzo church (in Florence) produced excellent results in terms of consolidation efficiency. In particular, the constancy of the consolidation efficiency over time was significantly higher than that exhibited by the $CaCO_3$ nanoparticles (applied here for comparison, always on the same type of Pietraforte sampled at the bill of the San Lorenzo church).

The choice of the $CaCO_3$ compound to perform a comparative study mainly depends on the texture and mineralogical composition of the Florentine Pietraforte sandstones, where Quartz ($SiO_2$) and Plagioclase ($NaAlSi_3O_8$—$CaAl_2Si_2O$) are the main components of the sandstone matrix and calcium carbonate represents the cementing element.

According to this consideration, the results shown after the application of the $CaCO_3$ nanoparticles directly on the Pietraforte sandstones highlight a satisfactory consolidating action but only immediately after the first application. Over time, the $CaCO_3$ nanoparticles wear out and undergo abrasion, resulting in a drastic decrease in consolidation efficiency. The Pietraforte treated with $CaCO_3$ becomes more and more permeable to water and much more resistant to the water vapor permeability, reducing the thermodynamic

exchanges with the external environment, fundamental for the natural survival of the Florentine sandstones.

As regards the comparison with the traditional and commercially available consolidant (i.e., Estel1000), the superiority of the $SiO_2$ nanoparticles is above all in the increased surface resistance, the increase in cohesion forces (tensile strength), the increase in the value of contact angle and improved resistance to perforation (i.e., drilling resistance also evaluated over time). Furthermore, the treatment based on $SiO_2$ nanoparticles guarantees the exchange with the water vapor of the environment, a necessary condition for the natural survival of the Florentine sandstone.

## 2. Experimental Section

### 2.1. Materials and Reagents for Synthesis

Methanol, ethanol, 1-propanol, ammonia solution, hydrochloric acid, 99% ethanol, and TEOS (tetraethyl orthosilicate) were purchased from (Sigma-Aldrich, Buchs, Switzerland). All other chemical reagents, especially the 1,4-butanediol (as working medium for $CaCO_3$ nanodispersion), were of analytical grade and used as received, without any purification. A Milli-Q water system was used to produce ultrapure water and all daily solutions, these latter for analytical measurements. For comparative study in terms of consolidation efficiency, $CaCO_3$ and the commercial Estel 1000 consolidating reagent have been selected. This choice lies in the fact that $CaCO_3$ is part of the Pietraforte matrix and also its main cement element, while the Estel 1000 represents a consolidating agent, widely applied on Florentine Pietraforte sandstones [16]. The latter consists of ethyl silicate in white spirit D40 solution. The SILO 111, on the other hand, is a ready-to-use water-repellent protector (not a consolidating agent) based on oligomeric organ siloxanes formulated at 10% in de-aromatized mineral white spirit. The RHODORSIL RC-80, formulated in organic solvent and based on ethyl silicates mixed with siloxanes, represents a water-repellent consolidant, but its applicability is significantly compromised by the low affinity toward the Florentine sandstone substrates. This is following the product data-sheet, which explains how this chemical regent is suitable for the restoration of natural stone and terracotta stone materials but is also recommended for the consolidation of porous stone supports. Florentine Pietraforte does not have these characteristics, and, therefore, this consolidation reagent (commercially available) is not suitable for the treatment of low porous Pietraforte sandstone.

Especially, $CaCO_3$ nanoparticles were synthesized at Tor Vergata University in Chemistry laboratories, according to our previous work [15], and the Estel 1000 agent was purchased by C.T.S. Srl (Altavilla Vicentina (VI)-ITALIA).

### 2.2. Synthesis of $SiO_2$ Nanoparticles

$SiO_2$ nanoparticles were fabricated at the Tor Vergata Chemistry Department laboratory by applying the modified Stöber method [29]. This modification to the conventional synthetic route can induce a large-scale mass production in a sustainable experimental procedure. Briefly, 0.6 mL of TEOS (tetraethyl orthosilicate) was hydrolyzed and condensed to give nanoparticles in an optimized mixed alcohol solution (i.e., methanol: ethanol was used as 8:1 *v/v*) with a basic catalyst of ammonia solution (1 mL of water and 3 mL of 30% $NH_4OH$ in 50 mL of total alcoholic solution). At the end of the synthesis, the $SiO_2$ nanoparticles were separated by centrifugation (for 13 min and working at 13,000 rpm) and washed three consecutive times with distilled water. Finally, 1 mg mL$^{-1}$ of nanoparticle dispersion was obtained using distilled water, suitable for Pietraforte consolidation treatments.

### 2.3. Preparation of $SiO_2$ Nanodispersion

$SiO_2$ nanoparticles were dispersed by sonication using a polytronic probe at 50 mW for 30 min at room temperature in distilled water as a working medium. The final aqueous concentration of 1 mg mL$^{-1}$ in distilled water resulted in the best working medium to prepare a stable nanodispersion over 6 months (stored at room temperature).

All the characterization techniques applied for the as-deposited $SiO_2$ Nanopowder and the corresponding liquid nanodispersion (realized in distilled water as a working medium) have been widely described in Electronic Supporting Information (see Supplementary Materials), also combined with a brief description of the experimental results, acquired on the collected samples.

### *2.4. Pietraforte Sandstone Collection, Consolidation Treatments and Characterization*
### 2.4.1. Samples Collection

The Pietraforte analyzed in this study (sample A, used here as control) was collected from the bell tower of San Lorenzo (in Florence) during its restoration, founded by The Italian Ministry for Cultural Heritage, Tourism, and by Opera Laurenziana, the restoration was completed in 2019. Samples A was collected by the restorers and conservator scientists belonging to the superintendency of the metropolitan city of Florence. One sample A (from which all the other samples for the experimentation were obtained) was collected on the upper level of the bell tower (see Figure 2), where a single block of ashlar stone was detached.

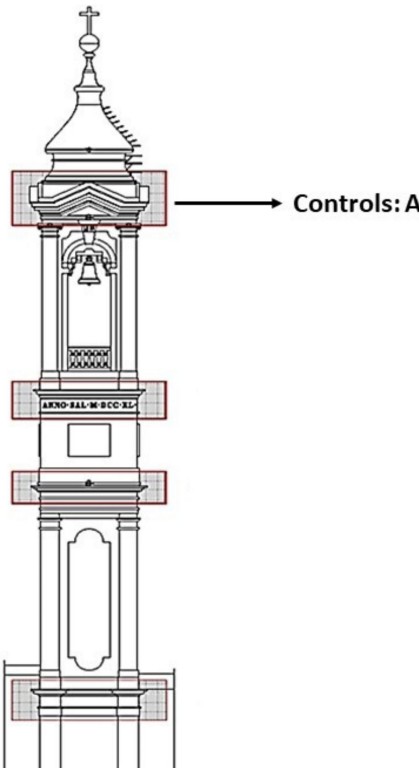

**Figure 2.** The bell tower of San Lorenzo and the corresponding sampling area, where a single block of Pietraforte was detached (which represented the control and the substrate from which all the samples used for the experimentation were taken).

One single sample A (without treatments) has 3 cm of thickness and an approximate surface area of 100–300 cm$^2$, and it was cut into several parts; one piece was untreated and was used as the control (sample A in Table 3), others were treated applying an aqueous dispersion of $SiO_2$ by capillary absorption (sample B on Table 3) or by brush (sample C on Table 3), respectively.

D and E samples both were treated by applying the $CaCO_3$ alcoholic (1,4-butanediol working medium) nano dispersion by capillary and brushing procedures, respectively. Finally, F and G were treated utilizing the commercial consolidating agent, called Estel 1000, by capillary and brushing applications, respectively.

**Table 3.** Sample dimensions, shape, and applied treatments.

| Samples | Dimensions (cm) and Shape | Treatments |
| :---: | :---: | :---: |
| A | $3 \times 10 \times 20$ cm$^3$ rectangular parallelepiped | No treatment (the control) |
| B | $5 \times 5 \times 10$ cm$^3$ prismatic samples | SiO$_2$-Capillary absorption |
| C | $5 \times 5 \times 5$ cm$^3$ cubic samples | SiO$_2$-Brushing treatment |
| D | $5 \times 5 \times 10$ cm$^3$ prismatic samples | CaCO$_3$-Capillary absorption |
| E | $5 \times 5 \times 5$ cm$^3$ cubic samples | CaCO$_3$-Brushing treatment |
| F | $5 \times 5 \times 10$ cm$^3$ prismatic samples | Estel 1000-Capillary absorption |
| G | $5 \times 5 \times 5$ cm$^3$ cubic samples | Estel 1000-Brushing absorption |

In Table 4, the main features of the only ashlar stone block that has detached itself from the upper level of the bell tower of San Lorenzo have been detailed and reported. The typical textural features of the Pietraforte sandstone having a low porosity were reported. In addition, the XRD pattern of the synthesized SiO$_2$ nanoparticles is presented in Figure S2A in Supplementary Materials. Briefly, the XRD pattern reveals a broad peak related to amorphous silica nanoparticles, at $2\theta = 23°$, with (101) Miller indices [30].

**Table 4.** Textural properties of Pietraforte control sample (the untreated sample A).

| Samples | $\gamma$ (g/cm$^3$) | $\gamma_s$ (g/cm$^3$) | Surface Area (m$^2$/g) | Total Pore Volume (cm$^3$/g) | $P_{tot}$ (%) | Total Porosity Decomposition (%) | | |
| :---: | :---: | :---: | :---: | :---: | :---: | :---: | :---: | :---: |
| | | | | | | Micro | Meso | Macro |
| Pietraforte control sample A | $2.71 \pm 0.01$ | $2.56 \pm 0.01$ | $7.2 \pm 0.5$ | 0.0120 | $5.70 \pm 0.14$ | $2.70 \pm 0.01$ | $3.00 \pm 0.01$ | —— |

$\gamma$ (g/cm$^3$); $\gamma_s$ (g/cm$^3$) and the Total Porosity decomposition (%) measurements exhibit an experimental error of 0.01. Surface area (m$^2$/g) and $P_{tot}$ (%) measurements show an error ranging from 0.5 to 0.14, respectively.

## 2.4.2. Consolidation Treatments of Pietraforte Samples

The consolidation treatments were carried out by applying two different approaches, the capillary adsorption and brushing treatment, respectively. Treated samples were left in the air under stable laboratory conditions for at least one month.

The capillary absorption was conducted as follows: $5 \times 5 \times 10$ cm$^3$ prismatic specimens of samples B, D, and F were used according to Ente Italiano di Normazione, UNI EN 15,801 (2010), [31]. Samples on a glass rod were positioned on the basal plate of a Petri dish containing the nanoparticle dispersion. Sample B base ($5 \times 5$ cm$^2$) came into contact with the dispersion volume. The consolidation absorption by capillarity lasted 3 h. In order to minimize solvent evaporation during the absorption procedure, the Petri dish and the Pietraforte sample were protected by a plastic container.

The brushing treatment was conducted as follows. Samples C, E, and G (four $5 \times 5 \times 5$ cm$^3$ cubic fragments) underwent surface saturation by nanoparticle dispersion application according to Ente Italiano di Normazione, UNI EN 15,801 (2010), [31]. Surface saturation was assessed as complete when the surface remained wet for one minute. The brushing treatment was carried out in a single application, followed by surface cleaning by a solvent-impregnated soft cloth to minimize deposit formation. After treatment, all samples were kept at room temperature for 24–48 h until reaching constant weight ($\pm 0.001$ g) and weighed after one month to estimate the nanodispersion absorbed by Pietraforte according to Ente Italiano di Normazione, UNI EN 15,802 (2010), [32].

The adsorbent amount of all products applied here on Pietraforte sandstone, and their corresponding penetration rate values have been summarized in Table 5.

**Table 5.** The adsorbed amount of SiO$_2$ nanodispersion and penetration rate on Pietraforte sandstones.

| Treatments | Adsorbent Amount of Products [a] [kg m$^{-2}$] | Penetration Rate [b] [mm/min$^{0.5}$] |
|---|---|---|
| Capillary SiO$_2$-application (sample B) | 6.0 ($\pm$0.02) | 7.6 |
| Brushing SiO$_2$-application (sample C) | 4.0 ($\pm$0.02) | 6.8 |
| Capillary CaCO$_3$-application (sample D) | 8.0 ($\pm$0.01) | 10.1 |
| Brushing CaCO$_3$-application (sample E) | 6.0 ($\pm$0.03) | 9.3 |
| Capillary Estel1000-application (sample F) | 3.2 ($\pm$0.04) | 4.2 |
| Brushing Estel1000-application (sample G) | 2.4 ($\pm$0.05) | 3.4 |

[a] The quantitative absorption of the consolidants was indirectly determined by dry weight measurements of the cylindrical samples (50 mm length, 15 mm diameter) before and after the treatment. The weight measurements were conducted after 6 weeks of storage at 23 °C/60% RH, for full curing of the consolidants. After 6 weeks the samples were dried in an oven at 40 °C until a constant mass was achieved and subsequently equilibrated at room temperature for one hour. [b] The consolidants were applied to sandstone by capillary adsorption; the stone specimens were partially immersed (3 mm depth) in the liquid consolidating agent for 3 h, respectively. The application time was determined by measuring the evolution of the capillary fringe (mm) on the lateral surface of the specimens, for 3 h. This latter is the time necessary for the wet fringe of the majority of the consolidants to reach the top of the specimens (3 cm). The space covered in the established time returns the penetration speed of the consolidating products.

The high penetration velocity values and the major amount of nanodispersion adsorbed by the stones resulted in greater in the case of the treatment by capillarity. The latter is not suitable for in situ applications, as in the case of brush treatment, which was therefore used for all subsequent measurements reported in this work.

For the characterization test (before and after the applications of the consolidating agents), the contact angle measurements, the surface hardness, the micro-drilling resistance, the tensile strength, the water adsorption coefficient, the resistance toward the Vapour diffusion, and the treatment efficiency (%) were fully evaluated, as described in the next paragraphs.

*2.5. Characterization Measurements*

2.5.1. Textural Properties

The specific surface area (Brunauer-Emmett-Teller, BET method) and total pore volume (by Gurvitsch (1915), [33]) were determined by adsorption/desorption of N2 at −196 °C using a 3Flex 3500 Micro metrics analyzer after sample outgassing at 200 °C for 2 h. The pore size distribution was determined by the Barrett-Joyner-Halenda (BJH) method (Barrett, Joyner, and Halenda 1951, [34]) from the adsorption isotherm. The uncertainty for the values of specific surface area was $\pm$0.5 m$^2$ g$^{-1}$. The porosity in the range 0.0037–150 µm (mesoporosity) and the relative pore size distribution were determined with the same apparatus cited above. The mesoporosity, together with the total open porosity, made it possible to calculate, as a difference, the microporosity (pores with radius $\leq$0.0037 µm), according to the classification of pore space proposed in [35–37].

2.5.2. Physical Properties

The physical characterization of the Pietraforte was performed on Samples A of size 1.5 × 1.5 × 3 cm$^3$. This latter was dried at 60 °C, and the dry weight W$_d$ was determined. The real volume V$_r$ and the bulk volume V$_b$ were determined using, respectively, a Quantachrome helium pycnometer and a Chandler Engineer mercury pycnometer. Then the samples were dipped into deionized water and weighed after saturation (constant wet weight W$_w$). With these data, the following parameters have been determined, according to the literature, [38]:

- real density ($\gamma$), = W$_d$/V$_r$;
- bulk density ($\gamma_s$) = W$_d$/V$_b$;

- total open porosity P% = $(V_b - V_r)/V_b \cdot 100$;
- water imbibition coefficient $IC_w$%, (expressed in weight) = $(W_w - W_d)/W_d \cdot 100$;
- water imbibition coefficient $IC_v$%, (expressed in volume) = $IC_w \cdot \gamma_s \cdot 100$;
- water saturation index SI% = $IC_v/P \cdot 100$.

### 2.5.3. Capillary Absorption and Contact Angle Determination

The static contact angle measurements to evaluate the hydrophobic characteristics of the treatment were performed using a PC-controlled NRL Rame-Hart apparatus, on control (untreated) and treated samples, according to Ente Italiano di Normazione, European Standards, EN 15,801 (2010) and UNI EN 15,802 (2010).

### 2.5.4. Effectiveness of Consolidation: Surface Hardness, Drilling Resistance Measurement Test, and Tensile Strength

Surface hardness was determined with a Martens sclerometer, equipped with a two-wheel handcart and a steel tip for scratching. Tests were carried out with the steel tip in contact with the specimen, ensuring it was perpendicular to the material surface. The upper surface of the handcart remained horizontal with the final force of 3 kgf applied on the tip. The handcart was subjected to a constant speed until reaching the set length. Every 3 mm along its length, the width of the incision was detected by applying a portable magnifying glass equipped with a light source and a micrometer, having a resolution of 0.02 mm. For each sample, 4–6 incisions were performed, and the results as the width of stroke (WS) are reported as the average.

The micro-drilling resistance was also evaluated using the drilling resistance test (DRMS) [39], presently considered the most suitable method for the evaluation of consolidation performance. For this investigation, 5 mm diameter tungsten drill bits have been applied, with a rotation speed of 400 rpm and penetration rate of 15 mm min$^{-1}$.

Increasing wear effect by drilling successive holes in an abrasive material (Pietraforte sandstone) was performed by using a Diaber drill bit $\varnothing$5 mm at 600 rpm and 10 mm/min advancing rate (as reported in the literature, [40]). Differential abrasion increase, in the case of two materials having different initial drilling resistance values, was carried out by a Diaber drill bit $\varnothing$5 mm.

Tensile strength was determined by ASTM C297/C297M (2016), [41].

### 2.5.5. Water Vapor Permeability ($P_v$%)

This quantity is calculated based on vapor permeability measurements, using the following formula:

$$P\,(\%) = \frac{(\vartheta_{NT} - \vartheta_T)}{\vartheta_{NT}} \times 100 \tag{1}$$

where: $\vartheta_{NT}$ represents the steady state steam flow of the untreated sample; $\vartheta_T$ represents the steady state steam flow of the treated sample, and finally, P (%) represents the decrease in vapor permeability following the application of the product [38].

### 2.5.6. Treatment Efficiency (%)

This additional parameter was calculated by Equation (2), reported below:

$$E_T(\%) = \frac{M_{NT} - M_T}{M_{NT}} \times 100 \tag{2}$$

where: $M_{NT}$ is the adsorbed water amount at t time by the untreated samples; $M_T$ is the adsorbed water amount at t time by the treated samples, and $E_T$ (%) represents the degree of treatment efficiency [38].

2.5.7. Water Adsorption Coefficient ($C_w$)

The capillary water absorption coefficient ($C_w$) was calculated according to EN1015-18 [42]:

$$C_w = 0.1 \, (M_2 - M_1) \tag{3}$$

where $M_2$ and $M_1$ are the mass of the specimen, in grams, after 90 min and 10 min of immersion, respectively. Prior to test, specimens were oven dried up to constant mass at 60 °C. Then, prismatic samples were immersed in deionized water for about 5 mm, and the mass variation was measured.

2.5.8. The Water Vapor Resistance Factor ($\mu$)

The water Vapour resistance factor ($\mu$) was measured according to the following equation:

$$\mu = \frac{\delta_A}{P_V} \tag{4}$$

where $\delta_A$ is air permeability ($1.94 \cdot 10^{-10}$ kg/(Pa m s)) in test conditions (20 °C and 50% RH) and Pv is water Vapour permeability [43].

2.5.9. Colorimetric Measurements to Evaluate Color Alteration

The optical appearance and modification of the treated stones were measured according to standard colorimetric methods [44]. An SP 820/830 spectrophotometer is used to measure the chromatic properties of the treated and untreated samples. Color values were detected in the CIE Lab space from the spectral reflectance factor of every pixel of the image. Mean values and standard deviations of L (lightness), a (redness), and b (yellowness) measurements from treated and untreated samples were used to obtain the average color difference DE. Chromatic variations in the CIE Lab space can also be represented by C and H parameter values as reported in the literature [45].

## 3. Results and Discussion

Briefly, the characterizations of the newly synthesized nanoparticles and of the nanodispersion that derives from them are reported in ESI (Figures S1–S5, see Supplementary Materials), following the order of presentation of the analytical techniques used for the characterization study. On the other hand, in this paragraph, the experimental results and their discussion regarding the study of the mechanical-engineering properties of Pietraforte, before and after the consolidation treatment performed by applying the new $SiO_2$ nanoparticles, have also been described.

We proceeded in this way because the novelty of the experimental work lies precisely in the improvement of consolidation efficiency in terms of the mechanical properties of the substrate. Following what has been stated, the first important result concerns the behavior of Pietraforte towards environmental water before and after the treatment with $SiO_2$ nanoparticles. In particular, the data shown in Table 6 demonstrated that the water imbibition coefficient $IC_v\%$ (expressed in volume) and the water saturation index SI% in the presence of $SiO_2$ treated Pietraforte sandstones were lower than those quantified for the untreated Pietraforte sandstone and the and that treated with Estel 1000, commercially available consolidating agent. This is reasonable considering the higher values measured for the total open porosity %, which increases in proportion to the decrease in the total real volume of the pores, due to the filling of the Pietraforte porosity, by the $SiO_2$ nanoparticles.

However, these results are less efficient than those obtained in the presence of the treatment with $CaCO_3$ nanoparticles because the latter represents a real cementitious element toward the Pietraforte sandstone consolidation activity. This cementitious action of $CaCO_3$ nanoparticles implies a significant decrease in the total pores real volume of the Pietraforte sandstones and, consequently, leads to an increase in the total open porosity (P%) and an equally significant decrease in the water absorption coefficient ($IC_v\%$), which is also strictly correlated to the saturation water index (SI%), see Table 6. These results were

very important because water represents one of the major damage problems of sandstones (and not only toward this kind of stone), mainly due to the phenomenon of capillary migration. As a result of the salt capillary transport, many electrolytes tend to recrystallize in areas where the phenomenon of water evaporation is greater, provoking mechanical stress and strain on the sandstones, which tend to disintegrate and detach.

**Table 6.** Textural features of Pietraforte sandstone and Adsorption coefficient, before and after treatments.

| Samples | Surface Area $(m^2/g)$ | Total Pore Volume $(cm^3/g)$ | $IC_w$ (%) | $IC_v$ (%) | $P_{tot}$ (%) | SI (%) |
|---|---|---|---|---|---|---|
| A (control) | $7.2 \pm 0.5$ | 0.0120 | $1.80 \pm 0.03$ | $4.60 \pm 0.07$ | $5.70 \pm 0.14$ | $83 \pm 1.51$ |
| C this work | $5.5 \pm 0.5$ | 0.0065 | $1.60 \pm 0.02$ | $4.40 \pm 0.05$ | $7.21 \pm 0.11$ | $61 \pm 2.01$ |
| E this work | $3.2 \pm 0.5$ | 0.0049 | $1.40 \pm 0.03$ | $4.20 \pm 0.01$ | $8.40 \pm 0.10$ | $50 \pm 2.32$ |
| G this work | $6.6 \pm 0.5$ | 0.0083 | $1.72 \pm 0.04$ | $4.52 \pm 0.07$ | $6.28 \pm 0.12$ | $72 \pm 1.86$ |

In both cases, the $SiO_2$ and $CaCO_3$ nanoparticles-based consolidation treatments perform better than the commercial Estel1000 treated Pietraforte sandstones and the untreated Pietraforte, respectively. The subsequent characterization in terms of cohesion forces and the static contact angle highlighted the better performances of the nanoparticles compared to the commercial agent and, between the two nanomaterials, the $SiO_2$ nanoparticles seem to give very performing results, even over time, especially when the drilling resistance forces have been measured (during time).

According to these considerations, all treated samples were characterized in terms of the static contact angle, the % increment of superficial hardness, the drilling resistance, and the tensile strength and compared with the control sample A, respectively. All data are summarized in Table 7.

**Table 7.** Characterization of untreated and treated Pietraforte samples in terms of cohesion forces.

| Sample | Contact Angle | Increment of Superficial Hardness | $DR_m$ [N] | Tensile Strength |
|---|---|---|---|---|
| | $(\vartheta \pm 3°)$ | (%) | l = drill bit $\varnothing$ 5 (mm) | (MPa) |
| A (control) | 48 | - | - | 13.6 ($\pm$ 1.2) |
| C this work | 80 | 35 | 34 | 29.8 ($\pm$ 2.2) |
| E this work | 159 | 80 | 36 | 36.7 ($\pm$ 2.2) |
| G this work | 49 | 22 | 27 | 14.5 ($\pm$ 2.3) |

The data in Table 7 would seem to show a better consolidation performance exhibited by $CaCO_3$ nanoparticles but in static conditions. When measurements, especially those carried out to evaluate the drilling resistance forces, were carried out over time, data showed that the Pietraforte treated with $CaCO_3$ nanomaterial was more subject to wear and abrasion phenomena compared to the same Pietraforte sandstone treated with the $SiO_2$ nanoparticles.

What has been said can be seen very well in Figure 3 of the text. In particular, Figure 3A shows the greater wear tendency of the Pietraforte treated with $CaCO_3$ nanoparticles compared to the sample treated with $SiO_2$ nanoparticles and the one treated with Estel1000 (commercial agent always based on ethyl silicate in white spirit D40 solution). In Figure 3B, we can see the greater abrasion of the Pietraforte treated with the $CaCO_3$ nanoparticles compared to that treated with the $SiO_2$ nanoparticles (and with the Estel1000, used as a commercial agent for comparative study). These results show the importance of the drilling resistance forces measurements over time because it highlights the greater degree of wear

and abrasion of the Pietraforte treated with $CaCO_3$, which is known to be an intrinsically less resistant material than silicon dioxide $SiO_2$ [46].

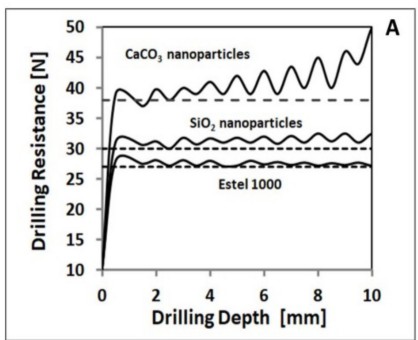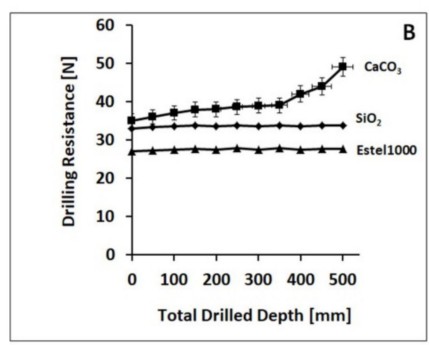

**Figure 3.** (**A**) Increasing wear effect by drilling successive holes in an abrasive material (Pietraforte sandstone), using a Diaber drill bit ∅ 5 mm at 600 rpm and 10 mm/min advancing rate (from Delgado Rodrigues and Costa 2004, [40]). (**B**) Differential abrasion increase in one Diaber drill bit ∅ 5 mm, measured on two materials having different initial drilling resistance values.

The measurement of the drilling resistance/[N] allows testing over time, a necessary condition to characterize materials that are subject to abrasion and wear. It is known in the literature [47] that $CaCO_3$ is less resistant than silicon dioxide ($SiO_2$) and other silicate-based compounds (such as Estel1000, which is a commercial agent based on ethyl silicate in white spirit D40 solution). What you notice is that with the same penetration (for example, 6 mm, as shown in Figure 3A), the wear forces expressed in Newton/[N] is almost constant in the case of $SiO_2$ nanoparticles and the commercial product Estel1000; while it presents/exhibits an increase of about 10% in the presence of $CaCO_3$ nanoparticles due to their greater wear and abrasion capacity. If the perforation in mm increases passing from 6 mm (as in Figure 3A) to 600 mm (as in Figure 3B), what is noticed is an increase in abrasion (always expressed in Newton/[N]) by another percentage factor of 10% was reached, in the presence of Pietraforte treated with $CaCO_3$ nanoparticles. The same trend is not observed in the case of Pietraforte treated with $SiO_2$ nanoparticles and/or consolidating agent (prepared in an organic solvent as a working medium), commercially available, as Estel1000. Under all these considerations, it is possible from this study to calculate a parameter that quantifies the efficiency of the treatments when different products have been applied (as in this study). This parameter, expressed as a percentage ($E_T$%) and evaluated according to the formula previously shown in the Experimental section, indicates the efficiency of the treatments, and it is highlighted in Table 8. In the same Table 8, the water permeability coefficient (P%) has been reported immediately after the first treatment and then after 6 months, which is really important for the comprehension of the shielding capacity of the Pietraforte after the treatment against capillary rising water, which is very deleterious for the substrate under study.

**Table 8.** Evaluation of the treatment Efficiency ($E_T$%) and the water permeability coefficient (P%).

| Treated Samples | $E_T$ (%) Immediately after Applying and Drying the Products | $E_T$ (%) after 6 Months from the First Application | P (%) Immediately after Applying and Drying the Products | P (%) after 6 Months from the First Application |
|---|---|---|---|---|
| A (control) | - | - | 30 | 30 |
| C | 86 | 86 | 34 | 34 |
| E | 95 | 63 | 58 | 55 |
| G | 70 | 70 | 57 | 57 |

The proposed $SiO_2$ nanoparticles seem to be the best nano consolidating agent for Pietraforte sandstone because of its high-efficiency parameter ($E_T\%$), which remains constant over time. Unlike, the $E_T$ (%) value calculated for the $CaCO_3$ nanoparticles resulted higher just after application but significantly decreased over time. This effect over time can be explained based on the tendency to wear and abrasion, typical of materials such as $CaCO_3$ (see Figure 3).

The best performances exhibited by $SiO_2$ nanoparticles also concern the water permeability coefficient (P%), whose values indicate the exchange between the sandstone and the external environment. If these values are high, this means that the treated rock, compared to the untreated one, opposes a greater resistance to vapor permeability with the negative result of altering the natural thermodynamic balances that are established between the rock and the external environment. Instead, minimum values of the vapor permeability coefficient indicate a predisposition of the rock to be permeated by the vapor to ensure the natural thermodynamic balances between the sandstone and the external environment.

In order to make this study exhaustive, it is rigorous to make a comparison with other treatments based on nanomaterials and commercial compounds of a silicate nature, applied mainly to the same type of Florentine sandstone (because, in this way, the comparison would be homogeneous). For this purpose, in Table 9, several different treatments applied to Florentine sandstones have been summarized, where mainly the water adsorption coefficient (%), the permeability to water vapor (P%) coefficient, and the tensile strength [MPa] were highlighted and compared with those exhibited by the sandstone control samples (without treatments).

**Table 9.** Comparative data analysis in terms of water adsorption coefficient, Water Vapour resistance, µ, and tensile strength for different consolidating treatments applied on Florentine Pietraforte sandstones.

| Samples | Water Adsorb. Coeff. ($C_w$) | µ | Tensile Strenght [MPa] | References |
|---|---|---|---|---|
| A (Control) | 2.5 | 27 | 13.6 | This work |
| C | 2.0 | 30 | 29.8 | This work |
| E | 1.8 | 50 | 36.7 | This work |
| G | 2.1 | 54 | 14.5 | This work |
| Pietra Serena (Control) | 2.2 | 29 | 8.2 | [48] |
| HAP-treated | 2.1 | 35 | 9.7 | [48] |
| TEOS-treated | 1.4 | 55 | 10.0 | [48] |
| Reference | n. r [a] | n. r. | n. r. | [49] |
| LWM10 | 32 | 13 | n. r. | [49] |
| LWM25 | 45 | 25 | n. r. | [49] |
| LWM50 | 61 | 48 | n. r. | [49] |
| GS Untreated (Control) | 4.4 | 18.8 | 4.1 (±0.9) | [48] |
| HAP-treated | 4.5 | 21.1 | 5.0 (±0.9) | [48] |
| TEOS-treated | 0.3 | 28.7 | 5.4 (±1.2) | [48] |
| PS Untreated (Control) | 2.2 | 29.3 | 8.2 (±2.2) | [48] |
| HAP-treated | 2.1 | 35.0 | 9.7 (±2.3) | [48] |
| TEOS-treated | 1.4 | 54.9 | 10.0 (±1.6) | [48] |

[a] not reported in the references [49], shown in Table 9.

According to these data, summarized in Table 9, the SiO$_2$ nanoparticles exhibit the best consolidating performances, especially in terms of the water adsorption coefficient, which resulted lower than those recorded in the presence of treatments based on Estel1000, HAP-treated sandstones, LWM10, LWM25, and LWM50. In all the other cases of the Table 9, for which the water absorption coefficient was lower than that shown by the SiO$_2$ nanoparticles, the μ parameter resulted in higher inhibiting the water vapor spontaneous diffusion from the external environment into the Florentine sandstone rocks. This aspect is negative for the consolidation treatment, as it alters the balance of natural exchange with the surrounding environment.

Furthermore, the best performance of SiO$_2$ nanoparticles in terms of Pietraforte consolidation was recorded by evaluating the tensile strength as a measurable quantity. In the case of SiO$_2$-based treatments, the tensile strength resulted higher compared to all the other values except for the CaCO$_3$ nanoparticles, which, however, present all the abrasion and wear problems previously described in the text.

Therefore, if a balance is made between all these measured quantities (contact angle, tensile strength, drilling resistance, water adsorbing coefficient, etc.) before and after the consolidation treatments, the one based on nanostructured SiO$_2$ results are the best ones. Indeed, over time the CaCO$_3$ nanoparticles wear and undergo abrasion resulting in greater water absorption effects (very dangerous for the conservation status of the Pietraforte sandstones).

Furthermore, all these excellent experimental results are also accompanied by the time factor, which is extremely reduced in the presence of SiO$_2$ nanoparticle-based treatments. Consider that, in a single week [50], the results summarized in the previous tables can already be highlighted, unlike the much longer application times, in the case of commercially available conventional consolidating agents. The latter exhibit an efficient consolidating action only after about 21 days from the first application [51] because they are used in solvent as a working medium, having viscosity and vapor pressure values, both too high when compared with those exhibited by aqueous nanodispersions, significantly lengthening consolidation times and efficiencies.

Under what has been stated, SiO$_2$ nanoparticles demonstrate a superior performance also in terms of optical properties (when compared with other nanoparticles and/or other commercially available consolidating agents), which will be discussed in the next paragraph. For a complete investigation aimed at establishing the goodness of the consolidating treatment, the study of the color coordinates was also carried out in this work, and the main results are reported in Table 10 and Table 11, respectively.

**Table 10.** Colorimetric L*, a*, b* data, total color difference ΔE* (±0.2), chroma difference ΔC* (±0.1), and hue difference ΔH* (±0.3) for A (the control sample, without treatments); B (treated by capillary adsorption) and C (treated by the brushing).

| Samples | L* | a* | b* | ΔE* | ΔC* | ΔH* |
|---------|-----|-----|-----|------|------|------|
| A (control) | 75.8 | 0.1 | 1.8 | —- | —- | —- |
| C this work | 75.7 | 0.1 | 1.8 | 0.1 | 0.1 | 0.1 |
| E this work | 76.0 | 0.2 | 1.9 | 0.2 | 0.2 | 0.1 |
| G this work | 77.2 | 0.3 | 2.0 | 3.2 | 0.9 | 0.3 |

**Table 11.** Aesthetical compatibility of SiO$_2$-based treatments.

| Risk of Incompatibility | Colour Difference | Treatments |
|-------------------------|-------------------|------------|
| Low | ΔE* < 3 | C and E samples |
| Medium | 3 < ΔE* < 5 | G sample |
| High | ΔE* > 5 | - |

The results demonstrate that no significant color differences $\Delta E^*$ [52,53] are observed for $SiO_2$ treatment applied on Pietraforte sandstone. The significant color difference $\Delta E^*$ has been observed in the case of Estel 1000 commercial silicate-based treatments. In the last case, the color change is not acceptable because the lightness and the other color components (especially a* and b* parameter) present a very large variation compared to the values detected for the untreated Pietraforte sandstone used here as control.

Data in Table 11 demonstrate that the best analytical performances have been reached by applying the $CaCO_3$ and $SiO_2$ nanoparticles on Pietraforte sandstones, while the Esteal 1000 in solvent treatment represents a medium risk level of the incompatibility for Pietraforte sandstone because of the $\Delta E^*$ value resulting in >3 units, as established in reference [54] and according to the Italian Guidelines for the restoration of stone buildings.

However, in the case of calcium carbonate, it is necessary to keep in mind the application procedure on sandstone, to avoid carbonation responsible for the whitening of the surfaces. There are several studies in the literature [55,56] and references cited therein, where the whitening effects provoked by the $CaCO_3$ nanoparticle applications are widely evident, as shown in Table 12. Furthermore, the authors of this study, in previous works, have found this, as highlighted in Figure S6 in the Supplementary Materials, where the presence of a bleaching event on the surface of the rock (i.e., the Florentine Pietraforte samples collected on the San Lorenzo's bill in Florence) treated with $CaCO_3$ nanoparticles, results quite evident.

**Table 12.** Comparison among the colorimetric $\Delta L^*$, $\Delta a^*$, $\Delta b^*$ data, total color difference $\Delta E^*$ for different stones treated with $CaCO_3$ nanoparticles, reported in the literature.

| Applied Treatment Materials | $\Delta L^*$ | $\Delta a^*$ | $\Delta b^*$ | $\Delta E$ | References |
|---|---|---|---|---|---|
| The samples treated with $CaCO_3$/Polymer nanocomposites | −1.21 | −0.34 | 0.33 | 1.30 | [55] |
| The samples treated with Ca(OH)$_2$/Polymer nanocomposites | 1.41 | −0.60 | −2.42 | 2.86 | [56] |
| The samples treated with Clay/Polymer nanocomposites | 1.68 | −0.22 | −1.09 | 2.01 | [56] |
| The samples treated with $SiO_2$/Polymer nanocomposites | 0.16 | −0.06 | −0.29 | 0.33 | [56] |

These results lead to the conclusion that the consolidation treatments based on nanoparticles do not seem to alter the optical properties of the Pietraforte, and among these, the best performance is obtained in the presence of $SiO_2$ nanoparticles (always produced at the Chemistry laboratories, University of Rome Tor Vergata).

Our data were in agreement with those recorded by several other different authors in the literature, according to which the realization of consolidating products based on nanocomposites of inorganic nanoparticles does not seem to alter the color properties of the treated substrates, which therefore do not undergo significant optical modifications.

In the light of all the results obtained, it can be said that the choice of synthesizing the new $SiO_2$ nanoparticles is perfectly in line with the efficiency of the consolidation treatment both in terms of a significant improvement in cohesion forces, a lower permeability to water and greater permeability to water vapor, a necessary condition for guaranteeing the natural exchange between the stone and the external environment. The lower resistance to water vapor permeation is a consequence of the constant drilling force over time (it remains the same for more than 6 months). This is the substantial difference with the $CaCO_3$ consolidating nanoparticles.

## 4. Conclusions

In this study, a new consolidating agent was synthesized in Chemistry Laboratories at Tor Vergata University of Rome (Italy), modifying an experimental procedure suitable to

produce a large-scale mass production of these new nanoparticles. These latter resulted in highly dispersible in water (thanks to the presence of hydrophilic functional groups such as Si-OH), completely avoiding the use of organic solvents, having a high impact on the end users and the ecosystem. The application of the new consolidating agent based on $SiO_2$ nanoparticles to the Pietraforte has important and different advantages if compared to our previously tested $CaCO_3$ nanoparticles. The new product has better performances in terms of resistance to perforation, wear, and abrasion even long range (for long times of exposure and consolidating exercise against Florentine sandstone), compared to the $CaCO_3$ nanoparticles (tested in our previous paper), which instead show excellent performance but only close to their first application. This means that over time, their resistance to drilling decreases, they wear much more easily (compared to $SiO_2$-treated sandstone), and tend to exhibit quite a significant surface abrasion phenomena. The experimental results highlight the better consolidation efficiency of the new consolidating agent on this kind of Florentine Pietraforte sandstone (having low porosity and a specific calcitic texture) in terms of water penetration protection, superficial cohesion forces, and an increase in surface resistance. The performance comparison of $SiO_2$ nanoparticles with commercial consolidants in solvents (such as Estel 1000, here tested) demonstrates two main concepts: (A) the restorative effects are obtained with a consolidation time over 1 week, significantly shorter when compared to the times of Estel 1000, exceeding 21 days; (B) $SiO_2$ nanoparticles perform better than Estel 1000 in terms of cohesion forces, also ensuring excellent preservation of the optical and color properties of the parent rock (without altering it after application).

Future studies are necessary to correlate all these measured chemical-physical quantities to develop an algorithm able to predict/estimate the efficiency of the new consolidating treatments.

**Supplementary Materials:** The following supporting information can be downloaded at: https://www.mdpi.com/article/10.3390/cryst12091182/s1. [1,32–37,57]

**Author Contributions:** Conceptualization, F.V. and P.P.; Data curation, F.V. and P.P.; Investigation, F.V., P.P., M.R., O.D., V.M., I.P. and S.D.A.; Writing—original draft, F.V., P.P., M.R. and O.D.; Writing—review and editing, V.M., I.P., G.S., G.F. and C.S.; Methodology, F.V. and P.P.; Funding acquisition, F.V. and P.P.; Resources, F.V. and P.P.; Supervision, F.V. and P.P. All authors have read and agreed to the published version of the manuscript.

**Funding:** This work was supported by Carlo Bugli Restauri Srl Company (Naples, Italy).

**Acknowledgments:** Authors wish to thank Soprintendenza Archeologia Belle Arti e Paesaggio per la Città metropolitana di Firenze e le province di Pistoia e Prato for the collection of the most representative Pietraforte samples, very useful for the characterization and experimentation of new nanomaterials. Moreover, Authors are very grateful to Arch. Fulvia Zeuli and Scelza Hosea (Soprintendenza Archeologia Belle Arti e Paesaggio per la Città metropolitana di Firenze e le province di Pistoia e Prato), for technical and logistical support in the restoration site, set up at the Bell Tower of the Monumental Complex in Florence (Italy).

**Conflicts of Interest:** The authors declare no conflict of interest.

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
