# Peer review of "SiO2 Nanoparticles as New Repairing Treatments toward the Pietraforte Sandstone in Florence Renaissance Buildings"

_crystals, doi:10.3390/cryst12091182_

Round 1

Reviewer 1 Report

This article is said to report a "new repairing treatment" using "SiO2 nanoparticles" for the "Pietraforte sandstone in Florence Renaissance buildings", and was proposed to Crystal's journal special issue "Archaeological Crystalline Materials". First of all, I struggle to understand where does it fit into the "Archaeological" part.
Nonetheless, there are some positives and negatives regarding the paper itself:

On the positive side, there is some potential in the approach and in exploring this new type of treatment for this stone in particular, and in the methods applied for the characterisation of the treatment's potential efficacy. The fact this is a less toxic material (aqueous suspension) and that may actually provide some cohesion is an important feature. The methodology is more or less standard and could indeed be used for the assessment that the authors wish to perform.

However, there are numerous negatives and I should point out several of my major and minor concerns:

1. Abstract
The abstract inside the pdf and the abstract in the submission platform are different, please double-check this.

The abstract inside the manuscript's pdf has some mistakes (Estel 100 instead of 1000, line 30), and fails to present clearly the novelty of the treatment.
Authors use the term "homemade" in line 18 and further on. This may not be the right term, please confirm it with a native speaker.

2. Introduction
The introduction is not written in a logical/systematic way.

2.1) Where is the short description/presentation of your crystalline material, its relevance and related conservation issues? ... There is something at the end of the introduction (lines 93-98) instead of the beginning, but is still extremely incomplete.

2.2) In the first paragraph (lines 39-42), the authors mention commercially available silicates for sandstones' consolidation, and in the next sentence specify alkoxysilanes and TEOS with 6 (seemingly, almost random) references:
Reference 1 is incorrectly formatted, check the last name of the authors (Xu and Li) 
Reference 2 has 0 mentions of sandstones
Reference 3 is, again, incorrect, authors and publication names are wrong...
Reference 5 has, once again, 0 mentions of sandstones
References 4 and 6 are the only ones actually enlightening, however, they apply to both commercial alkoxysilanes and lab-developed TEOS formulations in sandstones' consolidation.

2.3) In the second paragraph (lines 43-46), the authors mention that commercial products have a number of disadvantages related to the greater porosity of some types of Pietraforte... but all the way through the article is reported that Pietraforte has low porosity (see for instance lines 49, 147-148, 174, 197-198, 242).
What is the actual problem the authors wish to address? This should be clearly presented and referenced.

2.4) Many references are certainly missing, and several others do not represent what the authors state. For instance, ref 13 and 15 should be grouped together since these are merely examples of new nanocomposites developed recently, one using SiO2 as an added component. The same applies to ref 14, also merely an example. For example, you should explicitly state (see for instance references 13-14).
And where are exemplifying references for the application of consolidants to Pietraforte sandstones?
Where is the quick literature review to support the development of a new solution to a certain problem? What is the problem, after all? Again, this should be clearly presented.

2.5)  In lines 90-92, the authors state this study could "represent a new quantitative approach"... Please do inform on how you think that is possible through your study.

3. Experimental
Several details are missing.

3.1) Line 103 "All other chemical reagents" -- authors should specify which ones

3.2) Lines 106-107, why do you use only CaCO3 nanoparticles and Estel 1000 for comparison? You do not explain or fundament this choice anywhere in your paper.

3.3) Lines 127-128, the authors state that you nanodispersed SiO2 particles in distilled water, but in your ESI characterisation (Figure 1) they show results for the suspension in 1,4-butanediol. This should be clarified.

3.4) Moreover, lines 128-129, also concerning the results presented in ESI - if you present new data for this study as supplementary material, this should be preceded by the description of all the methodology and methods used in detail for the acquisition of that data so the readers can understand those results, instead of just pointing to ref 16.
Also, some short interpretation of the characterisation results should be provided even as ESI if the authors prefer. For instance, provide it for Figure 2, where the characterisation of the prepared nanoparticles is presented. One or two sentences will suffice.

3.5) From line 131 onwards, the detailed description of the samples collection is missing: where were the samples collected from exactly (which part of the bell tower?), when and by whom? What are their characteristics? Do the authors consider them to be new or aged samples?  This was not widely reported in the authors' previous work in ref 16, and even if some information can be found there, the authors should specify the samples' shape and size used in this study and a short summary of the characterisation results, properly referenced.

3.6) The denomination of the control and treated samples as well as the type of treatment should be clarified to the reader. Using Sample A, B and C and "treated by CaCO3" or "Estel" becomes very confusing, since B and C are the same product with different application methods. And I believe CaCO3 was also applied by the 2 different methods and the authors are comparing both results.
My advice is that you should add more columns/rows to your tables for the several treatments and types of application and redefine your denominations.

4. Results and discussion
Once more, systematisation is necessary.

4.1) Lines 164-165 "all measurements performed on Pietraforte". If so, please make the nomenclature in the 1st column of Table 2 become simpler. For instance: use "Pietraforte sandstone samples" as the title of the column and then the treatment and procedure in separate columns, if applicable. The table caption should also be more detailed.

4.2) Lines 167-174. Revise your English and the ideas you want to highlight, the text is sometimes very confusing.

4.3) Again in lines 177-179, if the authors have the XRD results from the previous study, please report briefly about the composition and textural characteristics of the Crystalline substrate in the Introduction or Experimental sections, supported by the ref 16 as the cited literature.

4.4) Table 3, line 5 -- The authors meant ref 16, not 6 I am sure.
The drilling resistance measurement in mm is confusing. Please clarify what your mean by the millimetres presented and how much was the increase.
At least one profile should be presented as a plot to provide a visual representation of a homogeneous (or not so homogenous) strength gain along the consolidated depth.

4.5) Lines 188-190: The authors should compare their results with more than one literature reference. Also, explain the advantage of a consolidation time in 1 week. This alone cannot be a decisive factor and I do not see any balance of advantages and disadvantages of each treatment considering the several parameters evaluated at the end of the discussion.

4.6) Place Fig 5 a and b side by side on the same page for continuous reading.

4.7) Lines 212-217 present an important point that should also be clarified in the introduction and abstract. However, the results presented in Table 4 do not point to the lack of success with the CaCO3 treatment. Could the authors demonstrate or reference works in which this occurs? Please provide them when stating something as in lines 227-229.

4.8) References 21, 21 in line 222 do not seem to fit.

4.9) For all the discussion and final conclusion (lines 230-234), please justify why Estel 1000 is chosen as the only commercial product for comparison in this case.

5. Conclusions
They seem too long for a short paper. Some of the thoughts could be presented at the end of the discussion and you could be more concise in your conclusive remarks.

Overall, there are large lacunae in this paper's presentation. 
- A summary information/characterisation of your crystalline substrate is missing
- A short review on the material (specifically Pietraforte sandstone) and the treatments applied in the past and present, their advantages and disadvantages, is missing in the introduction
- The systematisation of the problem and the proposed solution is missing
- The discussion on the advantages of a less toxic solution for the conservator's sake and the environment's sake is missing. Very timid mention of this in lines 203-205 and later on in the conclusions. This should be one of the most interesting features, if the treatment proves potentially successful.
- The justifications for all the decisions the authors take are missing
- A proper and open-minded discussion on the real potential of the presented treatment is missing, since the authors need to consider a broader scenario: long term efficiency, non-evaluated parameters and non-evaluated and non-compared treatments.

Therefore, I believe the authors should perform major revisions or resubmit the manuscript with a profound restructuration of their work, so it can be properly acknowledged, apprehended and valued by the interested communities.

Author Response

Answers to the Reviewers

Reviewer N.1 According to the useful suggestions and comments, all the answers to these latter have been summarized below. Our answers are reported in the red font! While, the tips, indications, suggestions, and advice provided by Reviewer No. 1, are shown in underlined black font.

  1. Questions about the abstract:

 The abstract inside the pdf and the abstract in the submission platform are different, please double-check this.

Answer: as requested, double-check was done and mistakes have been removed.

The abstract inside the manuscript's pdf has some mistakes (Estel 100 instead of 1000, line 30), and fails to present the novelty of the treatment.

Authors use the term "homemade" in line 18 and further on. This may not be the right term, please confirm it with a native speaker.

Answer: as requested we fixed the issues.

  1. Questions about introduction

The introduction is not written in a logical/systematic way.

2.1) Where are the short description/presentation of your crystalline material, its relevance, and related conservation issues? ... There is something at the end of the introduction (lines 93-98) instead of the beginning, but is still extremely incomplete. Answer: as requested, we presented the characteristics of the crystalline lattice of SiO2, in a more detailed and in-depth way in the introduction section, where the object of the work and its corresponding problems (related to its state of conservation) are firstly presented. The conservation status of the Florentine Pietraforte sandstones represents the aim of this experimental work.

2.2) In the first paragraph (lines 39-42), the authors mention commercially available silicates for sandstones' consolidation, and in the next sentence specify alkoxysilanes and TEOS with 6 (seemingly, almost random) references:

Reference 1 is incorrectly formatted, check the last name of the authors (Xu and Li).

Answer: as requested, we modified Reference 1.

Reference 2 has 0 mentions of sandstones.

Answer: We deleted Reference 2.

Reference 3 is, again, incorrect, authors and publication names are wrong...

Answer: We modified Reference 3, according to the Reviewer suggestion.

Reference 5 has, once again, 0 mentions of sandstones.

Answer: We deleted Reference 5.

References 4 and 6 are the only ones actually enlightening, however, they apply to both commercial alkoxysilanes and lab-developed TEOS formulations in sandstones' consolidation.

Answer: In agreement with the Reviewer’s advice and suggestions, we preserved references 4 and 6.

2.3) In the second paragraph (lines 43-46), the authors mention that commercial products have a number of disadvantages related to the greater porosity of some types of Pietraforte... but all the way through the article is reported that Pietraforte has low porosity (see for instance lines 49, 147-148, 174, 197-198, 242).

What is the actual problem the authors wish to address? This should be clearly presented and referenced.

Answer: The authors faithfully followed the suggestions of the Reviewer N. 1, uniquely defining the scientific subject of the publication. In this way, it is clearer for users to better understand the subject of the published paper.

2.4) Many references are certainly missing, and several others do not represent what the authors state. For instance, ref 13 and 15 should be grouped together since these are merely examples of new nanocomposites developed recently, one using SiO2 as an added component. The same applies to ref 14, also merely an example. For example, you should explicitly state (see for instance references 13-14).

And where are exemplifying references for the application of consolidants to Pietraforte sandstones?

Where is the quick literature review to support the development of a new solution to a certain problem? What is the problem, after all? Again, this should be clearly presented.

Answer: Following the indications of Reviewer N. 1, we have clarified and centered the problem that we want to investigate and present in this paper.

2.5)  In lines 90-92, the authors state this study could "represent a new quantitative approach"... Please do inform on how you think that is possible through your study. Answer: Considering the quantitative relationships between vapor permeability, resistance to water vapor diffusion, and treatment efficiency (all shown in the text), it is hoped to be able to correlate these quantities with others to formulate an innovative algorithm capable of quantitatively predicting the consolidating performance of one product instead of another.

Future experiments will be necessary, after this work, to join the drilling resistance (understood both as wear and abrasion of the materials) to the water absorption and to the permeability to water vapor coefficients, respectively. In this way, it would be possible to have a quantitative evaluation of the efficiency of the treatments, concerning and involving several stone samples, having different textures and porosities.

Experimental comments by reviewer N.1

  1. Experimental

Several details are missing.

3.1) Line 103 "All other chemical reagents" -- authors should specify which ones. Answer: Following the precious indications of Reviewer N. 1, the authors have integrated the missing information, adding all the chemical reagents and solvents applied in the experimental study.

3.2) Lines 106-107, why do you use only CaCO3 nanoparticles and Estel 1000 for comparison? You do not explain or fundament this choice anywhere in your paper. Answer: Following what was suggested by reviewer N.1, the authors explain and motivate the choice of only products, such as CaCO3 and Estel1000, to make a comparison with SiO2 nanoparticles (produced in the Chemistry laboratories of the University of Rome Tor Vergata). In particular, CaCO3 represents the typical cementitious binder of the Florentine Pietraforte sandstone while, Estel1000 is one of the commercial consolidants based on ethyl silicate mostly applied on the surfaces of monuments and historic residences, in the Tuscan region (from the Renaissance period and beyond). It is correct to mention the reason for the application of these two products, precisely the first time they appear in the body of the text, following what is underlined and highlighted by Reviewer N. 1.

3.3) Lines 127-128, the authors state that you Nano dispersed SiO2 particles in distilled water, but in your ESI characterizations (Figure 1) they show results for the suspension in 1,4-butanediol. This should be clarified.

3.4) Moreover, lines 128-129, also concerning the results presented in ESI - if you present new data for this study as supplementary material, this should be preceded by the description of all the methodology and methods used in detail for the acquisition of that data so the readers can understand those results, instead of just pointing to ref 16.

Also, some short interpretation of the characterisations results should be provided even as ESI if the authors prefer. For instance, provide it for Figure 2, where the characterisations of the prepared nanoparticles is presented. A few sentences will be needed to explain the experimental and the results, both presented in the ESI section. Answer: The recommendations of Reviewer # 1 in this paragraph are really important. Therefore, the Authors organize a short experiment with the details of the measurement and section procedures at the same time and prepare to comment for example Figure 2, albeit in short form. This contributes to making the scientific text of the manuscript more usable and understandable.

3.5) From line 131 onwards, the detailed description of the samples collection is missing: where were the samples collected from exactly (which part of the bell tower?), when and by whom? What are their characteristics? Do the authors consider them to be new or aged samples?  This was not widely reported in the authors' previous work in ref 16, and even if some information can be found there, the authors should specify the samples' shape and size used in this study and a short summary of the characterization results, properly referenced.

Answer: This aspect of sampling is very important and Reviewer N. 1 did very well to point out the deficiency. In fact, in this regard, it was specified (also with a new scheme of the San Lorenzo Bill Tower, introduced in the text of the manuscript), where the Florentine Pietraforte sandstone samples were collected, who took care of doing the sampling (i.e. the restorers and the conservation scientists) and finally the geometry and dimensions of the samples (strictly related to the cohesion forces experimental test, as the tensile strength; drilling resistance; increasing on the surface hardness; etc.;).

3.6) The denomination of the control and treated samples as well as the type of treatment should be clarified to the reader. Using Sample A, B and C and "treated by CaCO3" or "Estel" becomes very confusing, since B and C are the same product with different application methods. And I believe CaCO3 was also applied by the 2 different methods and the authors are comparing both results.

My advice is that you should add more columns/rows to your tables for the several treatments and types of application and redefine your denominations.

Answer: Also, in this case, Reviewer No. 1 rightly pointed out that in the presentation tables of the samples and the corresponding treatments to which they are subjected. There is no clarity for the readers who are interested in reading the manuscript. Thus the authors specified for each treatment (the one based on the SiO2 nanoparticles and those based on the two comparative treatments such as the CaCO3 and commercial Estel1000 procedures, respectively) how many samples were performed both by brush and by spray (respectively). The Control A sample is always the same for the entire duration of product testing.

  1. Results and discussion

Once more, systematization is necessary.

Answer: results were presented in a new format, the Authors presented the acquired data according to a systematic scheme, compatible with the preliminary premises/hypothesis of the work and self-consistent with the final results and their discussion.

Following the Reviewer N1 precious suggestions, the authors first presented the texture characteristics of the Pietraforte, before and after the treatments undergone. Subsequently, they presented the mechanical quantities related to the cohesion forces, discussing the results also as a function of the texture properties, previously reported. In this way, the differences with the CaCO3 nanoparticle-based treatment, previously published on the consolidation of Pietraforte, clearly emerged. To better understand the differences between the two types of nanoparticles and enhance the innovative aspect of the SiO2-based treatment, further physical-engineering quantities were evaluated with several experimental measurements, such as: drilling resistance as wear and abrasion features, respectively; the water adsorption coefficient; the water vapor permeability coefficient; etc.; From the latter, the advantage in the use and application (in both areas as the laboratory and in situ environment) of the new synthesized SiO2 nanoparticles (the real focus of this new paper), clearly emerged.

4.1) Lines 164-165 "all measurements performed on Pietraforte". If so, please make the nomenclature in the 1st column of Table 2 become simpler. For instance: use "Pietraforte sandstone samples" as the title of the column and then the treatment and procedure in separate columns, if applicable. The table caption should also be more detailed.

Answer: According to the useful suggestions of Reviewer N1, the authors organized in the first column of this Table 2 (that now, in the revised format of the paper, correspond to the final Table 1) the correct name of the Pietraforte sandstone samples and then, in the additional column, all the treatments and procedure were separately and distinctly reported. The captions are also shown in greater detail and this allows a more immediate reading of the text. Thanks to the suggestions and advice of Reviewer N1, even the tables can be used in a more orderly and clear way (especially regarding the quantitative parameters).

4.2) Lines 167-174. Revise your English and the ideas you want to highlight, the text is sometimes very confusing.

Answer: Following what was stated by Reviewer N. 1, the English language, the sentence construction, and the grammar have been deeply revised and corrected.

4.3) Again in lines 177-179, if the authors have the XRD results from the previous study, please report briefly about the composition and textural characteristics of the Crystalline substrate in the Introduction or Experimental sections, supported by the ref 16 as the cited literature.

Answer: According to Reviewer N 1 suggestions, the authors introduce the XRD results and their corresponding discussion in the Experimental section of the paper.

4.4) Table 3, line 5 -- The authors meant ref 16, not 6, I am sure.

Answer: Yes, Reviewer N. 1 is right what was meant was reference # 16 and not number 6. So, the authors correct it immediately.

The drilling resistance measurement in mm is confusing. Please clarify what your mean by the millimeters presented and how much was the increase.

At least one profile should be presented as a plot to provide a visual representation of a homogeneous (or not so homogenous) strength gain along the consolidated depth. Answer: the millimeters of drilling depth indicate how much the material wears and shortens, with the same force used to evaluate the resistance to perforation (i.e., the drilling resistance). Therefore, the Reviewer N 1 did well to point out the meaning of the millimeters shown in the table under the heading drilling resistance, and at the same time, he pointed out that a long-range evaluation is required to know the percentage variation of the drilling width and abrasion of the different materials.

So, according to Reviewer N. 1 suggestions, the authors clarify this concept by adding two plots, showing wear and abrasion over time. In his way, with the same reached drilling depth (for example 6mm), it is noted that the drilling resistance [N] remains almost constant in the case of the SiO2 nanoparticles and Estel1000 (this latter as a commercial product) while, an increase of 10% in terms of Drilling Resistance [N] for CaCO3 nanoparticles, is observed over time.

4.5) Lines 188-190: The authors should compare their results with more than one literature reference. Also, explain the advantage of a consolidation time in 1 week. This alone cannot be a decisive factor and I do not see any balance of advantages and disadvantages of each treatment considering the several parameters evaluated at the end of the discussion.

Answer: In the final Table 8, the authors report several different consolidating materials for similar Florentine sandstones, to compare their performances (especially in terms of water adsorption coefficient, Water Vapour resistance, μ, and tensile strength) with those exhibited by the synthesized SiO2 nanoparticles, proposed in this study for the first time. In this way, following Reviewer N. 1 suggestions, a better understanding of the performance of SiO2 nanoparticles is achieved, when compared with several other consolidating agents and not only with CaCO3 nanoparticles and the commercial product Estel1000, respectively.

Furthermore, these consolidation efficiencies manifest themselves already one week after the first application of the SiO2 nanodispersion. One of the great advantages of nanoparticles applied as consolidants, compared to traditional reagents (commercially available) is that not being applied in a solvent, they have a much faster action.

This effect depends on the spontaneous diffusion provided by the nanoparticles inside the porosities of the treated stone substrates. The same thing does not happen in the presence of commercially available consolidating agents, as a working solvent medium, because their efficiency and treatment yield depends very much on the evaporation rate of the solvents.

4.6) Place Fig 5 a and b side by side on the same page for continuous reading.

Answer: According to Reviewer N. 1, authors place the Fig 5 (that now corresponds to the new Fig 3) A and B side by side on the same page for continuous reading.

4.7) Lines 212-217 present an important point that should also be clarified in the introduction and abstract. However, the results presented in Table 4 do not point to the lack of success with the CaCO3 treatment. Could the authors demonstrate or reference works in which this occurs? Please provide them when stating something as in lines 227-229.

Answer: Following what was stated by the Reviewer N1, the authors explain in detail the lower performances exhibited by the CaCO3 nanoparticles compared to those shown by SiO2 nanodispersion, especially after the additional measurements of drilling resistance over time, where the greater tendency to wear and abrasion is demonstrated, especially in presence of CaCO3 nanoparticles (when applied to Florentine sandstone, such as Pietraforte). According to these considerations, the authors correct the Abstract, the Introduction text, and the lines 212-217; 227-229.

4.8) References 21, 21 in line 222 do not seem to fit. Answer: According to the suggestions provided by Reviewer N1, the authors changed reference 21 placed on line 222, by applying another one regarding the context in the full text.

4.9) For all the discussion and conclusion (lines 230-234), please justify why Estel 1000 is chosen as the only commercial product for comparison in this case.

Answer: The authors faithfully followed the indications of Reviewer N1, explaining and motivating the choice of Estel1000. The latter product based on ethyl silicate has always in the past been applied everywhere in the Tuscan region, in the residences and above all in the historic buildings of the Italian Renaissance. Its effectiveness over time as a consolidant has been amply demonstrated both with laboratory tests and field tests.

  1. Conclusions

They seem too long for a short paper. Some of the thoughts could be presented at the end of the discussion and you could be more concise in your conclusive remarks. Answer: The authors faithfully followed the indications of Reviewer N1, rewriting the conclusions again and summarizing only the main results obtained in the experimental work. In general, the authors reviewed the entire manuscript taking into account the observations summarized by Reviewer N.1, such as:

  • A summary characterization of your crystalline substrate has been introduced in the full text;
  • A short review of the material (specifically Pietraforte sandstone) and the treatments applied in the past and present, their advantages and disadvantages, is missing in the introduction and for this purpose, the authors have proceeded to integrate this section, as rightly suggested by the Reviewer N. 1
  • The systematization of the problem and the proposed solution is missing and for this purpose, the authors explain these two fundamental aspects for a clear understanding of the objective/goal of the work;
  • The discussion on the advantages of a less toxic solution for the conservator's sake and the environment's sake is missing. Very timid mention of this in lines 203-205 and later on in the conclusions. This should be one of the most interesting features if the treatment proves potentially successful. According to this, the authors explain very well the innovative aspect related to the aqueous nano dispersion application on Pietraforte sandstones, avoiding all the organic solvents (and organic mixtures) that do not blend well with the natural environment of the stones. The latter, it is known from the literature, interact thermodynamically with the open environment system, guaranteeing their stability, texture, and chemical-mineral composition over time.
  • The justifications for all the decisions the authors take are missing and for this purpose, the authors introduce all the necessary and sufficient reasons to make the reader understand the choices made and the achievement of the objectives set. For example, they explain why SiO2 nanoparticles are proposed as a new consolidant agent for the Florentine Pietraforte sandstones, also carrying out a comparative study with CaCO3 nanoparticles (applied on the same Pietraforte sandstone in a previous work published in the Analytical Letter 2021, as an international journal) and with a commercially produced available (widely used in the Tuscany region for the consolidation of historic Renaissance houses) such as the Estel1000.
  • A proper and open-minded discussion on the real potential of the presented treatment is missing since the authors need to consider a broader scenario: long-term efficiency, non-evaluated parameters, and non-evaluated and non-compared treatments. The authors were in full agreement with Reviewer N1 especially to validate the analytical performance of these new SiO2 nanoparticles over time. Therefore, new measures of cohesion forces have been introduced, in particular the mechanical quantity such as the resistance to perforation (i.e., drilling resistance) over time. This more in-depth study made it possible to clarify very well the differences between SiO2 and CaCO3 nanoparticles, especially in terms of the tendency to wear and abrasion manifested by calcium carbonate, compared to silicon dioxide. This allowed us to conclude that, in the operative choice of a conservative intervention to be carried out on the Florentine Pietraforte sandstone, it would be more appropriate to apply SiO2 nanoparticles given their greater resistance to wear.

The Abstract and the conclusions have been rewritten in a more summarized and orderly way!

     Signature                                                                                                                Date

Prof.ssa Federica Valentini                                                           Roma, on 25th July, 2022                                                                           

Reviewer 2 Report

The paper deals with SiO2 nanoparticles used as a new consolidating agent, on Pietraforte sandstone

The authors used SiO2 nanoparticles prepared in the lab and tested these nanoparticles, in agreement with standardized methods.

However, there are some aspects that should be corrected and added:

some figures have overlapping letters (Figure 1) 

some words ''repairing instead of reparing'' in the title

would be better to see some microscopic images of the wall tested before and after SiO2 application. 

The abstract should be reconsidered. The authors said: Nanoparticles (synthesized in large-scale mass production) have been characterized by 20 XRD-X Ray Diffraction, Raman and FTIR-Fourier Transform Infrared spectroscopy, and SEM-Scanning Electron Microscopy, but any of these investigation analysis are not presented in the paper.

The references list should be extended. Seems too minimalized.

Is not clear why the authors mentioned in the title SiO2 but in the paper text, they used SiO2 and CaCO3. Is not clear why. Also, how the authors could appreciate the compatibility between the monument wall (sandstone) with CaCO3.  the work seems written in a hurry, and without the necessary details for the readers.

I recommend to the authors to reconsider these aspects and resubmit the paper.

Author Response

Answers to the Reviewers

Reviewer N.2 According to the useful suggestions and comments, all the answers to these latter have been summarized below. Our answers are reported in red font! While, the tips, indications, suggestions and advices provided by the Reviewer No. 1, are shown in underlined black font.

Reviewer N 2. Some figures have overlapping letters (Figure 1).

Authors’ answer.  Following the Reviewer’s suggestions, the authors correct the Figure 1 about letter overlapping error.

Reviewer N 2. Some words ''repairing instead of reparing'' in the title.

Authors’ answer.  According to Reviewer N2, the Authors correct the title.

Reviewer N 2. Would be better to see some microscopic images of the wall tested before and after SiO2 application.

Authors’ answer. We provided picture to support our results in the supplementary information .

Reviewer N 2. The abstract should be reconsidered. The authors said: Nanoparticles (synthesized in large-scale mass production) have been characterized by XRD-X Ray Diffraction, Raman and FTIR-Fourier Transform Infrared spectroscopy, and SEM-Scanning Electron Microscopy, but any of these investigation analyses are not presented in the paper.

Authors’ answer. According to the Reviewer’s N2 suggestions and right comments, the Authors add all the Experimental sections and the results/data/discussion of the characterization study performed on these SiO2 nanoparticle samples, produced in Tor Vergata University at the Chemistry Laboratories (Roma). The authors decided to add this characterization information only in ESI section because the main goal of the new paper/manuscript concerns the cohesion forces study/investigation and the consolidation efficacy over time, after the innovative SiO2 nanoparticles-based treatment.

Reviewer N 2. The references list should be extended. Seems too minimalized.

Authors’ answer.  According to the Reviewer’s N2 suggestions, the Authors increase the References/Bibliography list, at the end of the revised paper/manuscript.

Reviewer N 2. Is not clear why the authors mentioned the title SiO2 but in the paper text, they used SiO2 and CaCO3. Is not clear why. Also, how the authors could appreciate the compatibility between the monument wall (sandstone) with CaCO3. The work seems written in a hurry, and without the necessary details for the readers.

I recommend to the authors to reconsider these aspects and resubmit the paper.

Authors’ answer.  Following what was stated by the Reviewer N2, the authors explain in the text (and here in the answer corresponding to the last question of the Reviewer), the reason for which a comparison is also made with the CaCO3 nanoparticles, already widely applied in the past on surfaces of Pietraforte Florentine sandstone.

CaCO3 represents the cement of the sandstone rock and therefore a comparison with other innovative materials could undoubtedly represent an interesting future challenge in the way of sustainable conservation of cultural assets (both outdoor and indoor), perhaps finding new systems with the same functionality but to lower environmental impact!

The Authors then evaluate the action of CaCO3 cement through and above all the characterizations that change the cohesion forces and the measurement of the contact angle. These characterizations allow establishing the affinity of the new material towards the damaged sandstone.

The Abstract and the conclusions have been rewritten in a more summarized and orderly way!

Date                                                                                                                         Signature

Roma, on 11th August 2022                                                               Prof. ssa  Federica Valentini    

Round 2

Reviewer 2 Report

I appreciate that the authors improved a lot the quality of this paper and this paper could be published in the present version.